# The Prognostic Value of the Hedgehog Signaling Pathway in Ovarian Cancer

**DOI:** 10.3390/ijms26125888

**Published:** 2025-06-19

**Authors:** Noor D. Salman, Lars C. Hanker, Balázs Győrffy, Áron Bartha, Louisa Proppe, Martin Götte

**Affiliations:** 1Department of Gynecology and Obstetrics, Münster University Hospital, 48149 Münster, Germany; noordawood.salman@ukmuenster.de (N.D.S.);; 2Department of Bioinformatics, Semmelweis University, 1085 Budapest, Hungary

**Keywords:** ovarian cancer, prognosis, hedgehog pathway, gene expression, TP53

## Abstract

The hedgehog pathway is a major regulator of cell growth and differentiation during embryogenesis and early development. The literature suggests that variations in this pathway’s genes play a role in tumor progression and response to therapy. This study aimed to assess the correlation between the expression levels of selected genes of this pathway and the progression-free and overall survival of ovarian cancer patients. Using the database Kaplan–Meier plotter, which includes the gene expression and survival data of 1565 ovarian cancer patients, higher expression levels of the genes *SHH*, *PTCH1*, *PTCH2*, and *GLI1* displayed better survival correlations, while *GLI*, *GLI3*, and *SUFU* correlated with adverse outcomes. Further dissection revealed a differential impact of the genes in specific clinical-histopathological categories. Notably, higher expression levels of *SUFU* were associated with a negative impact on ovarian cancer patients under many clinical–histopathological aspects. These results shed new light on the role of these genes in the chemoresponsiveness of ovarian cancer, especially *SUFU*, which could be considered a novel indicator for poor prognosis in epithelial ovarian cancer.

## 1. Introduction

Epithelial ovarian cancers are classified into two main types. Type I tumors, including genetically more stable low-grade serous, endometrioid, clear cell, and mucinous carcinomas, are slow-growing, often large but confined to the ovary, and arise from precursor lesions such as borderline tumors. Their origins vary: low-grade serous tumors come from the distal tubal epithelium, endometrioid and clear cell tumors from endometriosis, and mucinous/Brenner tumors from the transitional epithelium at the tubal-peritoneal junction. Type II tumors, most commonly the high-grade serous ovarian carcinomas (HGSOC), are aggressive, genetically unstable, often advanced at diagnosis, and typically arise from serous tubal intraepithelial carcinoma (STIC) in the fallopian tube, with frequent *TP53* mutations [1].

Due to difficulties in the early detection and treatment of epithelial ovarian cancer (OC) [2], the five-year overall survival rates are less than 50% [3]. Although molecularly targeted therapies, such as antiangiogenic agents, poly (ADP-ribose) polymerase (PARP) inhibitors, and folate receptor blockers, have been introduced, recurrence rates and mortality remain challenging [4]. Indeed, advancements are scarce in the detection of therapeutic targets in OC.

The hedgehog (HH) signaling pathway can be a potentially relevant modulator that can expand the diagnostic and therapeutic repertoire in malignant diseases. While HH signaling is inactive in mature mammalian tissues, it becomes active during tissue growth and survival in certain adult tissues [5]. Under abnormal conditions that lead to dysregulated HH signaling, these stem cells might eventually transform into cancer stem cells and trigger malignant transformation [6,7].

Observations suggest that HH signaling is active in human OC and is responsible for clonal growth and proliferation in OC cells [8]. The immunohistochemical expression levels of SHH, DHH, PTCH, SMO, and GLI1 proteins in ovarian tumors correlated strongly with malignant features, being higher in carcinomas than in borderline malignant tumors and, to a lesser extent, in benign cystadenomas. Additionally, this activation was not observed in normal ovarian surface epithelium [9]. However, the clinicopathological relevance of HH signaling in OC and its potential prognostic value are not clear at present.

Previous studies suggested that the activation of the Sonic Hedgehog signaling pathway is associated with poor prognosis in patients with several malignancies, including head and neck cancer, human glioma, bladder cancer, and small and non-small cell lung cancer [10,11,12], increasingly gaining attention as a therapeutic target [13], with a validated anticancer effect, as shown in basal cell carcinoma [14].

Building on prior research evaluating the HH pathway in other cancers, we aimed to similarly assess the relevance of the gene expression levels of key members and targets of the HH pathway to patient outcomes by assessing the progression-free and overall survival of OC patients. HH genes (*SHH*, *PTCH1*, *PTCH2*, *GLI1*, *GLI2*, *GLI3*, *HHAT*, and *SUFU*) were selected from the KEGG Database, with currently available oncological literature, as some of them have been the subject of previous research, especially the GLI family of transcriptional factors [15,16]. To identify the differential impact of the quantitative expression of each gene in OC on its outcome, the study also examined the prognostic value for the patient subgroups stratified by the histology, grade, and stage of the tumor, as well as the response to different therapeutic regimes, among other factors. The goal was to determine whether there is a differential impact of each gene of this pathway on the outcome of ovarian cancer, and a better understanding of the biological behavior of this cancer to pave the way for the implementation of individually targeted therapies for OC patients.

## 2. Results

### 2.1. General and Subset-Specific Correlation of HH Members’ Overexpression to the Survival Rates

To assess their potential prognostic value, we analyzed the correlation between the expression levels of the following eight components of the Hedgehog signaling pathway on the progression-free (PFS) and overall survival (OS) in ovarian cancer: *SHH*, *GLI1*, *GLI2*, *GLI3*, *PTCH1*, *PTCH2*, *HHAT*, and *SUFU*. A total of 1435 patients for PFS and 1656 patients for OS were analyzed using the KM plotter (Ovarian Cancer) database.

First, the impact of HH pathway constituents on PFS (Table 1, Figure 1), and secondly, their correlation with OS in ovarian cancer without any further classification were assessed (Table 1, Figure 2). We observed that *SHH* was correlated with better PFS (HR = 0.85; *p* = 0.0087), without affecting OS. In the same way, we noticed that *GLI1*, *PTCH1*, and *PTCH2* were correlated with better PFS and OS in OC (HR = <1, *p* = ≤ 0.05). On the contrary, higher levels of expression of *GLI2*, *GLI3*, and *SUFU* seemed to affect the PFS and OS adversely (HR > 1; *p*-value < 0.05). *HHAT* overexpression did not manifest any prognostic relevance.

Furthermore, the impact of HH members’ overexpression on the survival rates was assessed categorically in different histopathological and therapeutic subsets. The results are summarized in Table 2.

Except for SUFU, which was associated with poorer survival in serous ovarian cancer, no significant prognostic associations were observed for the target genes across different histological subtypes (Table 2-I). However, the validity of the analysis is limited since the vast majority of the cases in the dataset were of the serous subtype (1232 patients), and only a small number were of the endometrioid subtype (62 cases).

In the advanced stages of FIGO III and IV, *SHH*, *PTCH1*, and *PTCH2* overexpression are associated with better survival rates, while *SUFU, HHAT*, and *GLI3* demonstrated an adverse association with the outcome in OC patients.

*SUFU*, again, was the only member of the HH signaling pathway whose expression demonstrated an adverse prognostic effect on grade 3 OC with HRs of 1.38 and 1.4 and *p*-values of 6.3 × 10^−5^ and 0.007 for PFS and OS, respectively, as shown in Table 2-I.

Moreover, the higher expression levels of *HHAT*, *PTCH1*, and *SUFU* were associated with unfavorable PFS, OS, and PFS, respectively, in TP53-mutated OC as seen in Table 2-I.

*SHH*, *HHAT*, *PTCH1*, and *PTCH2* overexpression further improved the survival rates of ovarian cancer patients, particularly when optimal debulking was possible. Interestingly, even in cases where optimal debulking seemed impossible, *PTCH2* overexpression was associated with better OS. Once again, *SUFU* piqued our interest by displaying an adverse correlation with PFS, even in cases with optimal debulking, as summarized in Table 2-II.

To better understand the potential impact of the HH pathway gene expression on chemosensitivity as predictive biomarkers for treatment response, we analyzed its association with clinical outcomes in patients treated with platinum-based chemotherapy. Concerning progression-free survival (PFS), higher expression levels of *SHH*, *PTCH1* (Affymetrix ID BCNS), and *PTCH2* were associated with improved PFS, while higher expression levels of *GLI3* and *SUFU* correlated with worse PFS (Table 2-II, Figure 3). Regarding overall survival (OS), higher *PTCH2* expression was associated with improved OS, while elevated *SUFU* expression was linked to poorer OS (Table 2-II, Figure 4).

Notably, *SUFU* consistently demonstrated significant results across all categorical analyses, particularly within the clinically relevant subgroup (*n* = 139) of high-grade serous ovarian cancer at stages III and IV, even when the analysis was partly based on the GSE9891 dataset. Appendix A illustrates the adverse effect of *SUFU* expression on both progression-free survival (PFS) and overall survival (OS).

### 2.2. TNMplot-Based Comparative Analysis Reveals Gene Expression Differences Across Ovarian Tissue States

The expression profiles of the selected genes were assessed in normal ovarian tissues, as well as in tumorous and metastatic tissues from ovarian cancer by utilizing the public platform TNMplot (http://tnmplot.com, accessed on 7 April 2025) for comparative microarray gene expression data. The differential gene expression is outlined in box plots for comparison (Figure 5). The fold differences between tumorous and normal tissues, as well as metastatic and tumorous tissues, are indicated in Table 3.

As shown in Table 3, the expression levels of the genes *PTCH1*, *GLI2*, *GLI3*, and *HHAT* were significantly different among the three groups, while the differential expression levels of *SHH*, *PTCH2*, and *GLI1* in this comparative model were not significant (*p*-values > 0.02). Significant differences were observed in the expression levels of *GLI2* and *GLI3* when comparing tumorous to normal, with downregulation in their expression levels in tumors (FC 0.42 and 0.67, respectively). The two-group comparison (metastasis/tumor) revealed an almost two-fold upregulation of the *HHAT* gene. Interestingly, the expression level of *PTCH1* was downregulated in the metastatic analog.

### 2.3. Analysis of Gene Expression Levels in Primary and Metastatic Ovarian Cancer Cell Lines Derived from the CCLE Dataset

We next extended the gene expression analysis to a panel of 43 cell lines from the CCLE dataset, as it is still unclear whether the expression of these genes largely stems from the tumor cells or the cells in their microenvironment (e.g., cancer-associated fibroblasts, endothelial cells, pericytes, the immune cell infiltrate, etc.) [17]. Using the Mann–Whitney U test, none of the HH-related genes showed significant differential expression between metastatic and primary ovarian cancer cell lines (Table 4), which may be attributed to high inter-sample variability and the relatively small sample size. Notably, however, *PTCH1* and *GLI1* exhibited elevated expression levels in the primary ovarian cancer cell lines TOV112D and COV434. In addition, *PTCH1* expression was also elevated in the primary cell lines OVK18, TOV21G, OAW28, and JHOM2B. The heatmap below (Figure 6) illustrates the expression profiles of the 43 ovarian cancer cell lines analyzed.

### 2.4. Gene Expression Analysis in Cancer Cell Lines

As a random verification of the results retrieved from CCLE, the relative expression levels of *SHH*, *PTCH1*, *HHAT*, *GLI1*, *GLI2*, *GLI3*, and *SUFU* were determined using quantitative real-time PCR (qPCR) in two widely used OC cell lines; CAOV-3 as derived from primary site with *TP53* mutation and SKOV-3 derived from ovarian metastatic ascites, without *TP53* mutation. As illustrated in the box plots in Figure 7, *PTCH1*, *GLI3*, *HHAT*, and *SUFU* showed a higher mean expression level in both non-metastatic and metastatic cell lines, with an upward trend in the primary cell line CAOV3. While *SHH*, *PTCH2*, *GLI1*, and *GLI2* showed lower mean expression values when compared to above mentioned genes.

Comparing non-metastatic and metastatic ovarian cancer cell lines, the expression level of *SHH* and *PTCH2* was significantly higher in the primary line CAOV-3 (*p*-value < 0.05) than in the metastatic line SKOV-3, while the expression level of *GLI1* and *GLI2* was higher in the metastatic line SKOV-3, as demonstrated in Figure 7.

## 3. Discussion

### 3.1. Gene Expression Analysis Indicates HH-Related Genes as Potential Prognostic Markers in Ovarian Cancer

The HH signaling pathway is assumed to be primarily inactive in adult tissues, nevertheless, it helps regulate adult stem cells and is also involved in tissue maintenance and repair [5,18,19]. Other studies have exposed the mammalian ovary as a novel site of active HH signaling, suggesting that the HH signaling pathway remains active or is reactivated in follicles and oocytes [15,20]. Few studies have demonstrated the role of the HH pathway in human ovarian carcinoma, and even fewer have examined the prognostic role of the HH signaling molecules in ovarian cancer. However, they show conflicting results. Few studies used immunohistochemistry to investigate HH pathway activation [21,22,23], while others performed in situ hybridization and/or semiquantitative qRT-PCR [24]. The involvement of HH signaling in human cancers may be context-dependent, occurring in some tissues or cell lines but not in others.

In this analysis, we investigated the prognostic impact of gene expression levels of the HH pathway ligands and their transcriptional factors in a large patient cohort, divided into two groups—high and low expression, recruited under identical technical conditions, increasing the statistical reliability. The expression profile of a control sample with normal ovarian tissues was also compared. Independent of tumor characteristics, such as histology and grades, higher expressions of *SHH*, *PTCH1*, *PTCH2*, and *GLI1* have shown beneficial prognostic influence, whereas *GLI2*, *GLI3*, and *SUFU* are correlated with adverse clinical outcomes in OC patients. *HHAT* affected the outcome in OC only in certain subsets. The current study indicates a beneficial survival impact of *GLI1*, contradicting many previous studies, reporting *GLI1* as a potent manipulator in carcinogenesis and as a negative prognostic indicator in breast, gastric, and colorectal cancers [25,26,27]. Therefore, a coherent study design is required to scrutinize the controversial results regarding *GLI1* implications in OC. Meanwhile, the current finding was quite consistent with other studies in identifying *GLI2* as a predictor for poor clinical outcomes [24]. A previous study reported that *GLI1* and *GLI2* mRNA levels, indicators of active HH signaling, were significantly higher in cancer cells isolated from persistent/chemoresistant ovarian tumors than those isolated from matched primary tumors [28].

We further obtained differential subset-specific survival results. The current finding that the prognostic influence of HH signaling activation is not associated with any histological subtype of OC, except for *SUFU*, suggests that the morphological classification of ovarian cancer may not reflect the molecular pathogenesis of this disease. Similar to this study, Liao et al. found no significant correlation between the expression levels of *SHH*, *PTCH*, and *GLI1* and the grade and clinical features of the tumor [29]. Moreover, the co-occurrence of a mutated *TP53* gene, as a highly frequent event in the HGS-OvCa [30], was associated with adverse outcomes, not only with *SUFU* but also with *PTCH1* overexpression. *PTCH1* overexpression otherwise improved the survival rates of the overall cohort and in the subgroups of optimally debulked or platinum-treated patients. This suggests possible cross-talk or a corresponding drift in the signaling transactions of the HH pathway. Possible regulatory alterations in HH activity with the co-occurrence of the highly prevalent *TP53* mutation in OvCa are still unclear.

Higher expression levels of *SHH*, *PTCH1*, and *PTCH2* correlated with favorable prognosis in the OC subgroup treated with platinum-containing chemotherapeutic agents, which might indicate that an active HH status probably enhances chemosensitivity since higher levels of *PTCH1*, *PTCH2*, and *GLI1* are reliable indicators of an active HH status [31]. Furthermore, this study identified a distinctly similar impact of *GLI3* and *SUFU* on patients’ survival, which both exerted adverse effects in patients who received platinum-based chemotherapy, suggesting potential chemoresistance when *SUFU* or *GLI3* is highly expressed. Remarkably, *SUFU* consistently demonstrated a subset-specific adverse effect on survival outcomes across various clinico-pathological characteristics (serous histology, grade 3, FIGO III + IV stages, and mutated *TP53*), persistently worsening the prognosis in these categories, even with optimal surgical debulking and standard platinum-based chemotherapy.

The general agreement is that *GLI1* induces and *GLI3* represses HH target genes (*GLI1*, *PTCH1*, *Cyclin D1*, *c-Myc*, and *BCL-2*), whereas *GLI2* can act in either a positive or negative manner depending on post-transcriptional and post-translational processing events [32], which is in agreement with the major results of our study; probably due to the enhanced chemosensitivity when the HH pathway is active.

*SUFU* is a well-known negative regulator of the HH pathway, as it prevents *GLI* translocation into the nucleus [31]. The loss of *SUFU* results in the ligand-independent activation of HH signaling, indicating the central role of *SUFU* in the repression of this pathway [33]. Considering these facts and findings together, we can suggest that OC patients with a less active HH pathway probably encounter chemoresistance, which diminishes their chances of survival. Unfortunately, no previous studies have investigated the prognostic impact of *SUFU* in OC. Nevertheless, in medulloblastoma, a mutation in *SUFU* increases *SUFU* turnover, leading to sustained HH signaling activation, which is associated with the worst prognosis [34,35]. Ultimately, the consistently significant adverse survival correlation with higher levels of *SUFU* sheds new light on this biomarker, suggesting the need for an integrative study design that verifies the role of *SUFU* in OC.

### 3.2. Differential Involvement of HH-Related Genes in Tumorigenesis

Significant differences in the expression levels of *PTCH1*, *GLI2*, *GLI3*, and *HHAT* were calculated among normal, metastatic, and non-metastatic tumorous tissues by utilizing the public platform http://tnmplot.com for comparative microarray gene expression data. The two-group comparison—tumor/normal—showed, interestingly, a tumorous downregulation of the expression levels of *GLI2* and *GLI3*. In the metastatic/tumorous comparison, *PTCH1* showed a significant downregulation, whereas *HHAT* showed an almost two-fold upregulation of their expression levels in the metastatic tissues. These changes in the expression levels indicate the involvement of the respective genes of the HH pathway in tumor progression and its metastatic potential. We searched the literature that reports comparative expression levels of the respective genes. While no differential changes in *GLI1* were observed in our study, in another study, 56 primary advanced serous ovarian cancers and 12 normal ovarian tissues from postmenopausal women were immunohistochemically evaluated, and *GLI1* immunoreactivity was absent from the surface epithelium and stromal cells of the normal ovaries; however, a positive nuclear reaction was observed in 29% of the serous ovarian carcinomas [32]. Other studies have shown that *GLI1* overexpression promoted invasion and metastasis ability, as analyzed using RT-qPCR [36,37]. Similarly, expression assays using ovarian cancer cell lines and patient samples, as well as pooled normal ovarian samples subjected to RT-qPCR [8], showed downregulation of *PTCH1*, consistent with our results, and upregulation of *SHH* and *GLI1* expression levels compared to their levels in normal ovarian surface epithelium. *GLI2* displayed a decrease in the expression level in non-metastatic tumors compared to normal tissues, which may again reflect its inhibitory effects on HH pathway activation, assuming that an otherwise active HH pathway promotes carcinogenesis. Representatively, the relative gene expression levels of HH pathway genes in 16 primary grade 3 serous ovarian cancer samples standardized against normal ovarian cell lines demonstrated higher *GLI2* levels only in 25% of the samples [16]. Even in tumors with elevated expressions of the HH target genes *GLI1* and *PTCH1*, the expression of *SHH*, as a canonical driver of the HH pathway, is not necessarily high, suggesting other mechanisms of HH signaling activation in cancer [38]. Comparable studies that could further verify our results regarding the upregulated levels of *HHAT* in metastatic compared to non-metastatic tumors are, unfortunately, lacking.

### 3.3. Comparable Expression Levels of Hedgehog Pathway Genes in Ovarian Cell Lines Analyzed from CCLE Gene Expression Data

We could not show significant differences in the expression levels of our study genes between ovarian cancer lines of non-metastatic and metastatic origin. In the recent gene expression studies, efforts have been made to compare normal, benign, and non-metastatic cells as well as metastatic tissues of aggressive tumors. In most of these studies, it is not clear where the drift in gene expression occurs. In the context of short- and long-term survival in high-grade serous ovarian cancer, a recent study suggested that the transcriptomes of primary and metastatic tumors were similar to each other [39], however, the authors suggested that tumors from short-term survivors may be more clonal and genetically similar than tumors from long-term survivors, indicating an inherent resistance to treatment due to the genetic similarity between primary and metastatic tumors. As a conclusion here, the comparable gene expression levels among the 43 ovarian cancer lines can be principally accepted, assuming that momentary measurement of the gene expression levels does not necessarily reflect their ongoing interplay that orchestrates the following invasiveness and progression of the tumor, hence their prognostic implication.

### 3.4. Comparable HH Gene Expression in Epithelial Ovarian Cancer Cell Lines of Primary and Metastatic Origin, With and Without TP53 Mutation

We analyzed the mRNA expression profiles of HH-related genes—*SHH*, *HHAT*, *PTCH1*, *GLI1*, *GLI2*, *GLI3*, and *SUFU*—in two ovarian cancer cell lines. To the best of our knowledge, our study is the first work to comprehensively profile a wide range of HH pathway-related genes in primary vs. metastatic cell lines, comparing the findings with mRNA results from the online web tool http://tnmplot.com and the CCLE Gene Expression Data. Notably, the expression data of SKOV-3 and CAOV-3 from the CCLE dataset match our independent qPCR analysis data. However, a clear expression pattern of the genes was not found among these gene expression datasets.

A representative comparison of the gene expression profile between the primary cancer cell line with the *TP53* mutation (CAOV-3) and a metastatic cell line without the *TP53* mutation (SKOV-3) was made. A distinct expression pattern of these genes associated with a certain genetic mutation in these cancer cell lines (in this case, the *TP53* mutation) remains unknown and likely uncertain due to the variable genomic mutations present in the cell lines, as previously mentioned. In particular, we did not examine whether the mutations addressed in our cancer cell lines—the *TP53* mutation in CAOV-3, and the *CDKN2A* mutation in SKOV-3—possibly modulate the activation of the HH pathway. 

In another study, the inhibition of cell proliferation by suppressing HH signaling was observed in the following four cell lines with different *TP53* statuses in a TP53-independent manner: A2780 (wild-type *TP53*), A2780/CDDP (mutant vv*TP53*), SKOV3 (*TP53* deleted), and OVCAR3 (mutant *TP53*) [9]. This is relatively consistent with another report showing that G1 arrest through the inhibition of HH signaling was not affected by *TP53* status [40]. Ultimately, the heterogeneity in the expression levels of genes between microarray gene analysis and the cancer cell lines of the same histological type can be attributed to the extrinsic regulation of HH signaling in cancer. In this process, tumor cells activate the HH pathway in the non-malignant cells of their microenvironment by secreting HH ligand proteins [16]. This complexity may be due to differences in experimental systems (e.g., cell lines are prone to genotypic and phenotypic drift during their continual culture) and/or assay methods. It is also conceivable that they might reflect the heterogeneity of autocrine and paracrine signaling in OC [32]. In a study on human epithelial OC cell lines (SKOV-3, CAOV-3, and others) and 12 human OC tissue specimens, tumor-derived SHH has been shown to stimulate stromal vascular epidermal growth factor (VEGF-C) secreted by cancer-associated fibroblasts (CAFs) to promote lymphangiogenesis in OC via the HH/VEGF-C signaling axis [21]. Recent evidence indicates that the sonic HH pathway is often recruited to stimulate the growth of cancer stem cells (CSCs) and orchestrate the reprogramming of cancer cells via the epithelial-mesenchymal transition (EMT) [41].

Differences in *GLI1* and *SUFU* gene expression levels among control tissue, borderline tumors, and carcinomas have been reported by other studies. *SUFU* was detected with lower expression levels in higher FIGO (International Federation of Gynaecology and Obstetrics) stages compared to lower stages [42], as the authors confirmed that the HH signaling pathway was active in ovarian tumors and that *GLI1* and *SUFU* were associated with the tumor type and FIGO stage. It seems that the activation occurs downstream of the membrane components of the pathway, suggesting non-canonical activation, especially considering that simultaneous overexpression of *GLI1* and *SUFU* under classical physiological circumstances is not typical.

Targeting the HH pathway may hold promise as a novel approach for treatment, widening the landscape of ovarian cancer therapy. In another study, cyclopamine, a natural inhibitor of the HH pathway, and its semi-synthetic derivative IPI-926 demonstrated significant antitumor activity when administered after conventional chemotherapy in primary tumor xenografts. They mediated their effects in both stromal and epithelial cells by significantly reducing GLI1 expression in the stroma and tumor cells [36], although previously, canonical inhibition of the HH pathway could not be clinically confirmed [43]. A recent study found that *GLI1* transcriptionally regulates FANCD2 expression, a tumor suppressor gene, and a key member of the FA-BRCA and the homologous recombination (HR) pathways, implicating the importance of GLI1-dependent expression of FANCD2 genes in tumorigenesis and chemoresistance in OC [44]. Inhibition of GLI1 in OC reduced FANCD2 expression, creating a state of HR deficiency, where PARP inhibition is known to be more effective. This study addresses the rationale that GLI1 depletion or inhibition can enhance the effectiveness of PARP inhibitors in BRCA-proficient patients, as they are the largest group in OC.

To overcome resistance to canonical inhibitors of the HH pathway, it is necessary to design methods that can operate beyond the canonical state of Hedgehog pathway activation in ovarian cancer.

There were several limitations to our study. One key limitation is that the survival datasets included the mRNA levels of our study genes rather than the levels of the encoded protein as the final functional effector. However, during protein synthesis, substantial regulatory processes can occur after mRNA is produced, which may occur beyond the transcriptional and translational levels, with different protein turnover rates [45]. Therefore, it is undoubtedly important to validate these findings at the corresponding protein levels. Nevertheless, microarray-based studies focused on creating genomic biomarkers to classify and predict cancer survival have been an accepted routine practice for over 20 years in developing and enhancing the care of cancer patients [46].

Another limitation of this analysis relevant to TNMplot is the inability to distinguish between different histological subtypes of epithelial ovarian cancer when comparing tumor and normal samples. Given the distinct cells of origin and molecular characteristics of each subtype—particularly the dominance of p53-mutated high-grade serous carcinomas arising from fallopian tube epithelium—grouping all tumor types together may obscure subtype-specific gene expression patterns. Therefore, while TNMplot is valuable for preliminary assessments, findings should be interpreted with caution in light of these constraints.

Besides, an important limitation of this study is that some of the ovarian cancer cell lines used were derived from tumor cells isolated from ascitic fluid (EFO-21, OVCAR-3, ONCO-DG-1, SKOV-3, OC-314, and HEYA8), rather than from solid peritoneal metastases. While ascites-derived tumor cells reflect advanced disease, they do not fully recapitulate the biological behavior and microenvironmental context of solid metastatic implants. Therefore, gene expression or phenotypic characteristics observed in these cell lines may not entirely represent the metastatic progression or tissue-specific interactions seen in solid peritoneal lesions. This distinction should be considered when interpreting the translational relevance of in vitro findings. Our qPCR-based and CCLE dataset cell line analysis could be useful for choosing an in vitro model to study the relevant signaling pathways; however, for this setting, the mutational status of the gene and the expression of the relevant proteins should be analyzed in addition.

## 4. Materials and Methods

### 4.1. Kaplan–Meier (KM) Plotter Tool for Survival Assessment

This study focused on retrieving gene-related survival estimations in OC patients from the publicly accessible Kaplan-Meier Plotter web tool (https://kmplot.com, accessed on 7 April 2025). This tool is based on mRNA expression data and survival data from the Gene Expression Omnibus (GEO) and The Cancer Genome Atlas (TCGA), which have been integrated for analysis. To avoid differences due to different sensitivity, specificity, and dynamic range in detecting gene expression levels for specific genes by different technologies, the search has been narrowed to only tumor samples examined using the in situ oligonucleotide array platforms GPL96 (Affymetrix Human Genome U133A Array), GPL571 (Affymetrix Human Genome U133A 2.0 Array), and GPL570 (Affymetrix Human Genome U133 Plus 2.0 Array). The oligonucleotide gene array files were MAS5 normalized. Then, a second scaling normalization was executed to set the mean expression in each array to 1000. Only the probes present in the GPL96 platform were used in the scaling normalization to prevent platform-specific differences due to the higher probe number in the GPL570 arrays [47,48].

According to the quantile expressions of a specific gene, the patients’ collective data were divided into two groups using the web interface to analyze the prognostic value of that gene. These groups were then compared using progression-free survival (*n* = 1436) and overall survival (*n* = 1656) data. A KM survival plot was mapped, and the significance was calculated. This integrative data analysis tool was used to investigate the prognostic value of the genes involved in the HH signaling pathway and their transcriptional products (*SHH*, *PTCH1*, *PTCH2*, *GLI1*, *GLI2*, *GLI3*, *HHAT*, and *SUFU*). The Affymetrix IDs that represent our genes are as follows: *SHH*: 207586_at, *HHAT*: 219687_at, *GLI1*: 206646_at, *GLI2*: 228537_at, *GLI3*: 227376_at, *PTCH1*: 209816_at, *PTCH2*: 221292_at, and *SUFU*: 222749_at. Only the JetSet best probe set was chosen; this set selects the optimal probe set for each gene using a scoring method established to assess each probe set for specificity, coverage, and degradation [47].

Before running the analysis, the patient groups were filtered based on stage, histology, grade, TP53 mutation status (mutated or wild-type), and treatment parameters, including debulking status and the applied chemotherapy. The progression-free survival and overall survival rates in each group were investigated. The clinical properties and their proportional distribution in the cohorts are summarized in Table 5, Table 6 and Table 7.

The Hazard ratio (HR) with a 95% CI and the log-rank *p*-value were calculated in each KM survival plot, with the log-rank *p*-value cutoff defined as <0.05. The KM plot shows the association between the investigated marker and survival, where the samples are grouped according to the median (or upper or lower quartile) expression of the selected gene. The median expression was used as the cutoff to categorize the samples into high- and low-expression cohorts in each analysis. The false discovery rate (FDR) was computed using the brainwaver library in R, version 1.6, as integrated into the KM plotter tool.

### 4.2. Comparative Online Gene Expression Analysis Using TNM Plots

To assess the expression level of the study genes among normal, non-metastatic, and metastatic ovarian tissue, we resorted to the public platform TNMplot in the comparative microarray gene expression data in normal ovarian tissues as well as in tumorous and metastatic tissues from ovarian cancer, where the study genes have been investigated for expression profiles among the above mentioned tissue types and tested for fold differences as a comparison between tumorous and normal, as well as metastatic and tumorous tissues and outlined in box plots [49]. The significance in gene expression between the three groups is provided by using the Kruskal–Wallis method with a *p*-value set to <0.02. In order to compare the groups with each other, the differential expression is represented by fold change (FC).

### 4.3. Analysis of Ovarian Cell Line Gene Expression Data from the CCLE Datasets

In another layer of assessing the expression level of the study genes in non-metastatic and metastatic ovarian cancer, we searched the Cancer Cell Line Encyclopedia (CCLE) datasets. Gene expression data from 1019 cell lines were retrieved from the DepMap portal [50]. From these, 43 ovarian cancer cell lines have been identified, comprising 13 metastatic, 29 primary tumor-derived, and one immortalized ovarian cell line. TPM-normalized expression values were used to generate heatmaps, constructed using the ComplexHeatmap R package Bioconductor version: Development (3.22) [51]. For statistical comparison between primary and metastatic cell lines, we performed the Mann–Whitney U test, with significance set at *p* < 0.05.

### 4.4. Cell Culture and Quantitative Real-Time PCR Analysis

Concurrently, and as a verification of the results retrieved from CCLE, the expression levels of our target genes were assessed in two human ovarian cancer cell lines, acquired from the American Type Culture Collection (ATCC/LGC), Promochem (Wesel, Germany), featured by varying degrees of genetic complexity and mutation. One cell line was derived from the primary site (CAOV-3), displaying a TP53 mutation, and another one was derived from ovarian metastatic ascites (SKOV-3), with no TP53 mutation. Following the product manual, total RNA was isolated from the cultured cell lines using the innuPREP RNA Mini Kit (Biometra, Göttingen, Germany). One microgram of the total isolated cellular RNA was used for reverse transcription into cDNA using the First Strand cDNA Synthesis Kit (Thermo Fisher Scientific, Schwerte, Germany), random hexamer primers, and M-MuLV reverse transcriptase. The target gene expression levels were determined using real-time reverse transcriptase–polymerase chain reaction (RT-PCR). The gene expression levels were measured using an ABI PRISM 7300 Sequence Detection System (Thermo Fisher Scientific, Schwerte, Germany) using SYBR Green dye (Thermo Fisher Scientific) and the cycling conditions recommended by the manufacturer. The cycle threshold (Ct) values from each sample were normalized to their corresponding *β-actin*. Ct melting curve analysis was performed to confirm specific product amplification. The expression levels were calculated using the 2^−∆∆Ct^ formula relative to the housekeeping gene *ß-actin* as an endogenous standard. All experimental samples were analyzed in triplicate, derived from three independent experiments. Student’s *t*-test was used to test the significance of differential gene expression levels in each of the two cell lines. All tests were two-sided, and a *p*-value of less than 0.05 was considered significant. The results were demonstrated graphically in box plots.

## 5. Conclusions

In conclusion, it can be validly stated that the activation of the HH pathway affects the prognosis of ovarian cancer. This analysis of survival data in such a substantially large cohort provides, for the first time, evidence that the overexpression of *SHH*, *PTCH1*, *PTCH2*, and *GLI1*—known indicators of active HH signaling—improves progression-free and overall survival, while *GLI3* and *SUFU* worsen progression-free and overall survival in epithelial, particularly serous, ovarian carcinomas. The findings are sufficiently broad to suggest, for the first time, *SUFU* as a potential indicator for poor prognosis in ovarian cancer and a sensitive predictive marker for resistance to platinum-based chemotherapy, highlighting the need for an alternative approach to treatment for the affected patients. As a future direction, a thorough evaluation of SUFU and GLI1 as prognostic biomarkers of high sensitivity and specificity in independent discovery and validation cohorts appears worthwhile. Moreover, our data mark SUFU as a potential therapeutic target that could be evaluated in future preclinical studies.

## Figures and Tables

**Figure 1 ijms-26-05888-f001:**
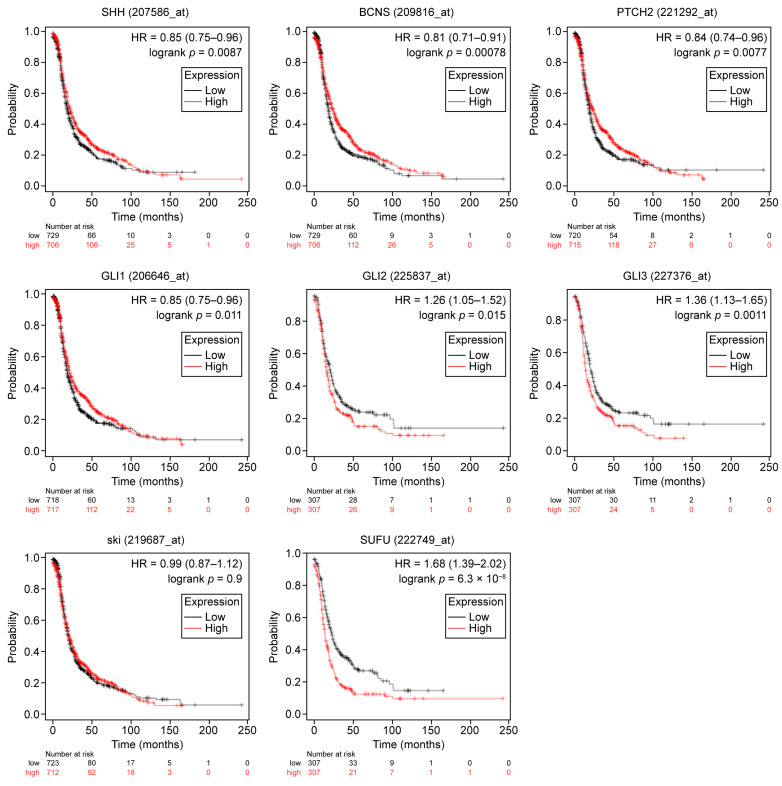
Prognostic significance of HH-signaling-related gene expression for progression-free survival in ovarian cancer. The patient cohort was divided into two groups: low (black) and high (red) gene expression, according to their respective median gene expressions. The patients were censored at the threshold. Survival is displayed as Kaplan–Meier curves; the computed HR, 95% CI, and log-rank *p*-values are given. SHH (207586_at); PTCH1 = BCNS (209816_at); PTCH2 (221292_at); GLI1 (206646_at); GLI2 (228537_at); GLI3 (227376_at); HHAT = ski (219687_at); SUFU: 222749_at; HH: Hedgehog; HR: Hazard ratio.

**Figure 2 ijms-26-05888-f002:**
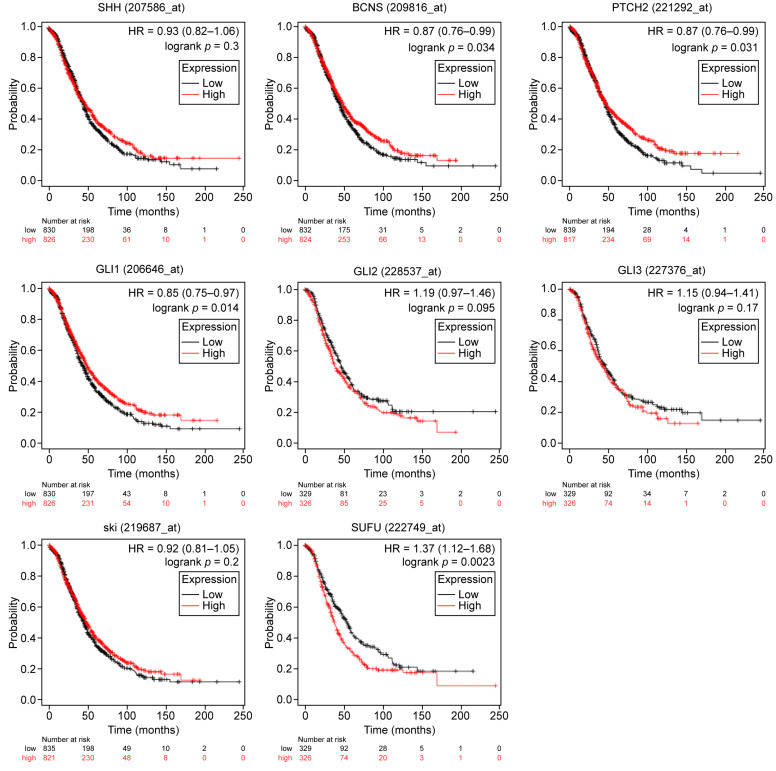
Prognostic significance of HH-signaling-related gene expressions for overall survival in OC. The patient cohort was divided into two groups: low (black) and high (red) gene expression, according to their respective median gene expression. The patients were censored at the threshold. Survival is displayed as Kaplan-Meier curves; the computed HR, 95% CI, and log-rank *p*-values are given. SHH (207586_at); PTCH1 = BCNS (209816_at); PTCH2 (221292_at); GLI1 (206646_at); GLI2 (228537_at); GLI3 (227376_at); HHAT = ski (219687_at); SUFU: 222749_at. HH: Hedgehog; HR: Hazard ratio.

**Figure 3 ijms-26-05888-f003:**
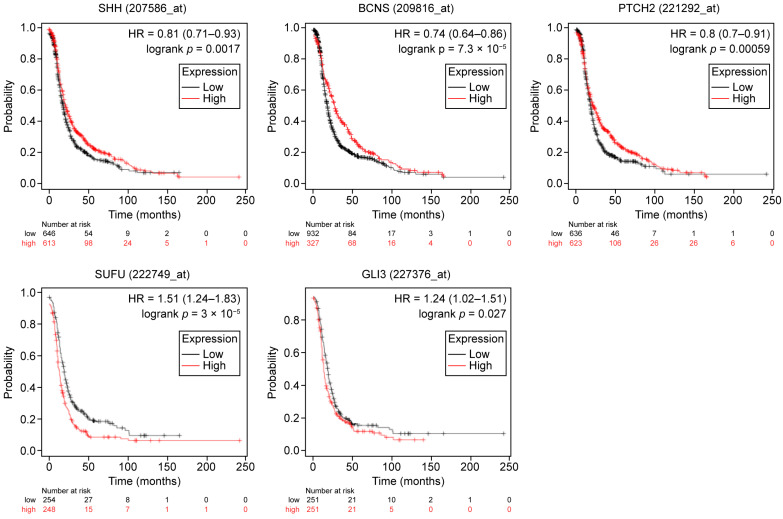
Progression-free survival as impacted by the expression levels of the studied HH pathway genes in OC patients (*n* = 1259) who received platinum-based chemotherapy. Log-rank *p*-values and hazard ratios (HRs; 95% confidence interval) are shown. HH: Hedgehog; HR: Hazard ratio.

**Figure 4 ijms-26-05888-f004:**
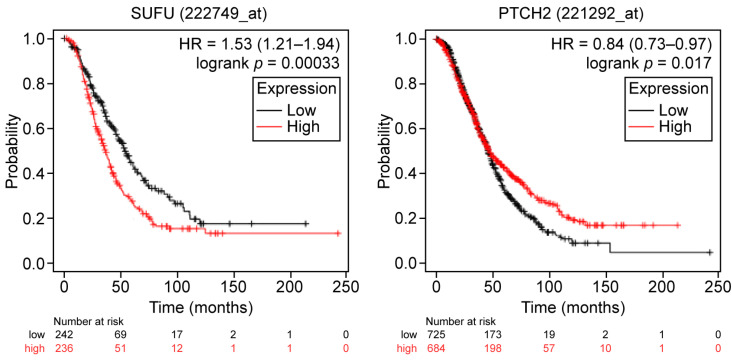
Prognostic correlations of the expression levels of the studied HH pathway genes in OC patients (*n* = 1409) who received platinum-based chemotherapy. Overall survival curves are plotted. Log-rank *p*-values and hazard ratios (HRs; 95% confidence interval) are shown. HH: Hedgehog; HR: Hazard ratio.

**Figure 5 ijms-26-05888-f005:**
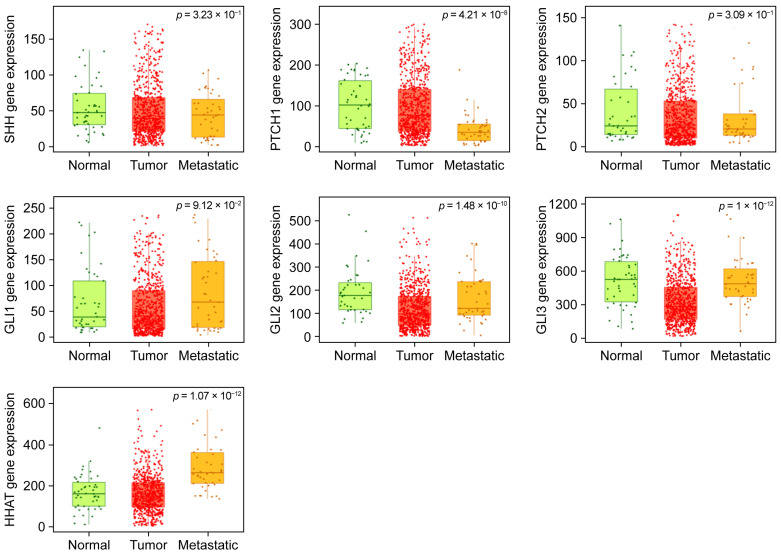
Boxplots of comparative expression profiles of *SHH*, *PTCH1*, *PTCH2*, *GLI1*, *GLI2*, *GLI3*, and *HHAT* applied to gene array data of normal, malignant, and metastatic ovarian tissues. The analysis platform allowed for the simultaneous comparison of tumor, normal, and metastatic data using the Kruskal–Wallis test.

**Figure 6 ijms-26-05888-f006:**
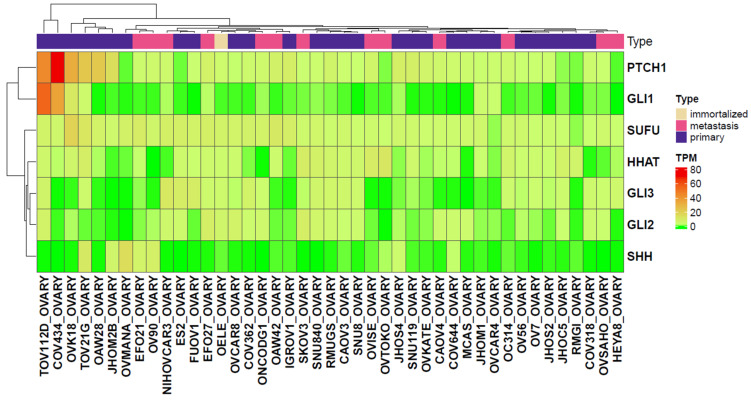
Heatmap of the raw TPM values of the selected HH-related genes using data from CCLE. The x-axis represents the cell lines, while the y-axis represents the genes. TPM: Transcripts per million.

**Figure 7 ijms-26-05888-f007:**
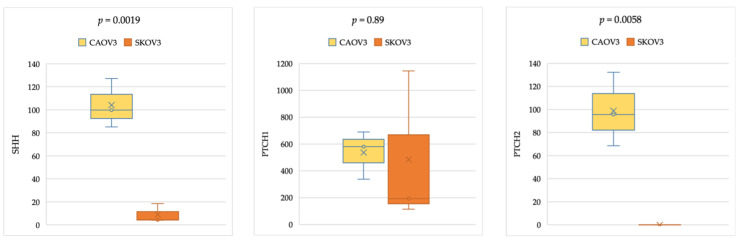
Relative expression levels of HH signaling pathway-related genes in the primary vs. metastatic ovarian cancer cell lines CAOV3 and SKOV-3. The relative expression levels of the *SHH*, *PTCH1, HHAT, GLI1, GLI2, GLI3*, and *SUFU* genes were quantified using qRT-PCR in two ovarian cancer cell lines representative of serous cystadenocarcinoma. Individual experiments were normalized to β-actin, and the relative expression levels are represented as 1000 × 2^−ΔCt^. The data represent the mean ± SEM (standard error of the mean). HH: Hedgehog; CAOV3: derived from primary site; SKOV-3: derived from ovarian metastasis.

**Table 1 ijms-26-05888-t001:** Prognostic significance of HH signaling pathway-related genes for progression-free survival and overall survival in the ovarian cancer cohort as a whole. HR (hazard ratio) and *p*-values of the following eight components of the HH signaling pathway: *SHH*, *GLI1*, *GLI2*, *GLI3*, *PTCH2*, *SUFU*, and *HHAT*. HH: Hedgehog.

Gene	Affymetrix ID	No.	Progression-Free Survival		No.	Overall Survival	
			HR	*p*-Value		HR	*p*-Value
*SHH*	207586_at	1435	0.85	0.0087	1656	0.93	0.3
*PTCH1*	209816_at	1435	0.81	0.00078	1656	0.87	0.034
*PTCH2*	221292_at	1435	0.84	0.00077	1656	0.87	0.031
*GLI1*	206646_at	1435	0.85	0.011	1656	0.85	0.014
*GLI2*	228537_at	614	1.26	0.015	655	1.19	0.095
*GLI3*	227376_at	614	1.36	0.0011	655	1.15	0.17
*SUFU*	222749_at	614	1.68	6.3 × 10^−8^	655	1.37	0.0023
*HHAT*	219687_at	1435	0.99	0.9	1656	0.92	0.2

**Table 2 ijms-26-05888-t002:** Subset-specific correlation of HH members’ overexpression to the survival rates; I histopathological parameters; II therapeutic parameters; n.s. (not significant). The quoted results represent the *p*-value/HR. HH: Hedgehog; HR: Hazard ratio.

**I**	**Histology (Serous)**	**Grade 3**	**Mutated TP53**	**Stage III + IV**
*SHH*	n.s.	n.s.	n.s.	OS 0.03/0.85
*HHAT*	n.s.	n.s.	PFS 0.01/1.33	OS 3.6 × 10^−7^/1.7
*PTCH1*	n.s.	n.s.	OS 0.04/1.26	OS 0.01/0.8
*PTCH2*	n.s.	n.s.	n.s.	OS 0.003/0.72
*GLI1*	n.s.	n.s.	n.s.	OS 0.035/0.85
*GLI2*	n.s.	n.s.	n.s.	n.s.
*GLI3*	n.s.	n.s.	n.s.	PFS 0.005/1.4
*SUFU*	PFS 5.0 × 10^−7^/1.7 OS 5.0 × 10^−4^/1.5	PFS 6.3 × 10^−5^/1.38 OS 0.007/1.4	PFS 0.02/1.56	PFS 6.3 × 10^−6^/1.59 OS 1.37/0.006
**II**	**Optimal Debulking**	**Suboptimal Debulking**	**Platinum Chemotherapy**	
*SHH*	PFS 0.001/0.7	n.s.	PFS 0.001/0.8	
*HHAT*	OS 0.01/0.78	n.s.	n.s.	
*PTCH1*	PFS 2.1 × 10^−5^/0.6 OS 0.015/0.8	n.s.	PFS 0.0004/0.08	
*PTCH2*	OS 0.0005/0.7	OS 0.0004/0.7	PFS 0.0006/0.8 OS 0.017/0.8	
*GLI1*	n.s.	n.s.	n.s.	
*GLI2*	n.s.	n.s.	n.s.	
*GLI3*	n.s.	n.s.	PFS 0.005/1.4	
*SUFU*	PFS 0.002/1.66 OS 0.05/1.5	PFS 0.013/1.4 OS 0.03/1.4	PFS 3 × 10^−5^/1.5 OS 0.0003/1.5	

**Table 3 ijms-26-05888-t003:** Differential gene expression in normal, tumorous, and metastatic ovarian cancer tissues. Fold change (FC) in expression levels of two comparison groups (tumor vs. normal, metastasis vs. tumor).

Gene	*p*-Value	Tumor/Normal Fold Change *p*-Value		Metastatic/Tumor Fold Change *p*-Value	
*SHH*	3.2 × 10^−1^	1.03	2.01 × 10^−1^	0.71	6.86 × 10^−2^
*PTCH1*	4.2 × 10^−8^	1.07	2.67 × 10^−1^	0.37	1.45 × 10^−6^
*PTCH2*	3.1 × 10^−1^	0.73	1.36 × 10^−1^	0.67	6.31 × 10^−2^
*GLI1*	9.1 × 10^−2^	0.76	7.59 × 10^−2^	1.2	4.04 × 10^−1^
*GLI2*	1.5 × 10^−10^	0.42	6.95 × 10^−10^	1.41	2.51 × 10^−2^
*GLI3*	1 × 10^−12^	0.67	1.84 × 10^−7^	1.65	2.97 × 10^−1^
*HHAT*	1.0 × 10^−12^	0.97	2.30 × 10^−1^	1.95	4.21 × 10^−7^

**Table 4 ijms-26-05888-t004:** Differential gene expression levels between primary cell lines and metastatic cell lines.

GeneID	*p*-Value	Mean in Primary	Mean in Metastatic	Median in Primary	Median in Metastatic	Fold Change
*HHAT*	0.93	3.21	3.82	2.76	2.14	1.19
*GLI2*	0.75	2.26	2.91	1.71	1.71	1.29
*GLI3*	0.93	2.63	2.48	1.93	1.97	0.94
*SUFU*	0.63	5.47	5.71	5.02	4.50	1.04
*GLI1*	0.97	4.15	0.65	0.40	0.34	0.16
*SHH*	0.97	1.37	1.37	0.10	0.10	1.00
*PTCH1*	0.87	9.89	4.02	3.23	4.30	0.41

**Table 5 ijms-26-05888-t005:** The clinical characteristics of ovarian cancer patients in the microarray datasets used in the analysis.

GEO ID	GEO Platform	No. of Samples in Dataset	Death Event	Median Overall Survival	Serous/Endometrioid and Others	Grade (1/2/3)	Stage (1/2/3/4)	Optimal Debulking	Platinum-Based Chemotherapy
GSE14764—available online: https://www.ncbi.nlm.nih.gov/geo/query/acc.cgi?acc=GSE14764 (accessed on 7 April 2025).	GPL96	80	21	33.7	68/7	NA	8/1/69/2	27	78
GSE15622—available online: https://www.ncbi.nlm.nih.gov/geo/query/acc.cgi?acc=GSE15622 (accessed on 7 April 2025).	GPL571	35	28	37.8	31/0	0/7/28	0/0/26/9	NA	20
GSE19829—available online: https://www.ncbi.nlm.nih.gov/geo/query/acc.cgi?acc=GSE19829 (accessed on 7 April 2025).	GPL570	28	17	43.0	NA	NA	NA	NA	NA
GSE3149—available online: https://www.ncbi.nlm.nih.gov/geo/query/acc.cgi?acc=GSE3149 (accessed on 7 April 2025).	GPL96	116	68	48.5	NA	NA	NA	NA	NA
GSE9891—available online: https://www.ncbi.nlm.nih.gov/geo/query/acc.cgi?acc=GSE9891 (accessed on 7 April 2025).	GPL570	285	110	31.0	264/20	19/97/164	24/18/217/22	160	242
GSE18520—available online: https://www.ncbi.nlm.nih.gov/geo/query/acc.cgi?acc=GSE18520 (accessed on 7 April 2025).	GPL570	53	41	40.4	NA	NA	NA	NA	NA
GSE26712—available online: https://www.ncbi.nlm.nih.gov/geo/query/acc.cgi?acc=GSE26712 (accessed on 7 April 2025).	GPL96	185	129	47.6	NA	NA	NA	90	184
GSE23554—available online: https://www.ncbi.nlm.nih.gov/geo/query/acc.cgi?acc=GSE23554 (accessed on 7 April 2025).	GPL96	28	14	49.3	NA	NA	NA	NA	28
GSE26193—available online: https://www.ncbi.nlm.nih.gov/geo/query/acc.cgi?acc=GSE26193 (accessed on 7 April 2025).	GPL570	107	76	49.9	79/8	7/33/67	20/11/59/17	38	93
GSE27651—available online: https://www.ncbi.nlm.nih.gov/geo/query/acc.cgi?acc=GSE27651 (accessed on 7 April 2025).	GPL570	39	25	57.5	NA	NA	NA	NA	NA
GSE30161—available online: https://www.ncbi.nlm.nih.gov/geo/query/acc.cgi?acc=GSE30161 (accessed on 7 April 2025).	GPL570	58	36	45.8	44/1	2/19/29	0/0/49/5	23	58
GSE32062—available online: https://www.ncbi.nlm.nih.gov/geo/query/acc.cgi?acc=GSE32062 (accessed on 7 April 2025).	GPL570/GPL6480	10	7	70.3	10/0	0/0/10	All advanced	NA	NA
GSE63885—available online: https://www.ncbi.nlm.nih.gov/geo/query/acc.cgi?acc=GSE63885 (accessed on 7 April 2025).	GPL570	75	66	42.7	70/1	0/9/48	0/2/63/10	15	75
TCGA—available online: http://cancergenome.nih.gov/(accessed on 7 April 2025).	GPL3921	557	292	33.3	522/0	4/65/440	14/25/400/81	340	505
	Total	1656		39.1	1122/51	56/325/1024 ∗	107/72/1079/189	801	1409/1438 *

NA: data not available. The asterisk indicates the actual number of patients at the public platform Kaplan–Meier Plotter who have OS data and only arrays that passed the quality control. The numbers represent the no. of samples the analysis has run on after selecting our setting parameters on the page interface referred to.

**Table 6 ijms-26-05888-t006:** Proportional distribution of characteristics in the ovarian cancer microarray database.

Parameter	Cohort	Proportion of Patients
Platform	GPL96 (Affymetrix HG-U133A),	25%
	GPL570 (Affymetrix HG-U133 Plus 2.0)	46.7%
	GPL571/GPL3921 (Affymetrix HG-U133A 2.0)	2%
	GPL3921	34%
	GPL6480	NA
Histology	Serous	67.7%
	Endometrioid/others	3%
Grading	1	3.3%
	2	19.6%
	3	61.7%
Stage	1	6%
	2	4%
	3	65%
	4	11%
Surgical debulking	Optimal (residual tumor <1 cm)	48%
	Suboptimal	NA
Chemotherapy	Platinum-based	77%
OS	Follow-up time (months)	39.1
	Proportion of events (death)	56%

NA: date not available. These percentages represent the minimum values, if available, due to a lack of clinical specifications in some of the datasets.

**Table 7 ijms-26-05888-t007:** Collective and subset-related sample numbers used in the survival analysis applied in the Kaplan–Meier plotter online tool. The first number represents the sample size tested for PFS, and the second number, following the forward slash, represents the sample size analyzed for OS. PFS: Progression-free survival; OS: Overall survival.

	SHH PFS/OS	HHAT PFS/OS	PTCH1 PFS/OS	PTCH2 PFS/OS	GLI1 PFS/OS	GLI2 PFS/OS	GLI3 PFS/OS	SUFU PFS/OS
Serous	1104/1207	483/523	1104/1207	1104/1207	1104/1207	483/523	483/523	483/523
Endometrioid	51/37	44/30	51/37	51/37	51/37	44/30	44/30	44/30
G 1 + 2	293/380	189/203	293/380	293/380	293/380	189/203	189/203	189/203
G 3	837/1015	315/392	837/1015	837/1015	837/1015	315/392	315/392	315/392
Stage I + II	163/135	115/83	163/135	163/135	163/135	115/83	115/83	115/83
Stages III + IV	1081/1220	494/487	1081/1220	1081/1220	1081/1220	494/487	494/487	494/487
mTP53	483/506	124/124	483/506	483/506	483/506	124/124	124/124	124/124
Opt. Debulking	696/801	240/243	696/801	696/801	696/801	240/243	240/243	240/243
Subop. Debulking	459/536	234/235	459/536	459/536	459/536	234/235	234/235	234/235
Platinum	1259/1409	502/478	1259/1409	1259/1409	1259/1409	502/478	502/478	502/478

## Data Availability

The data are contained within the article.

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
