# Peer review of "The Prognostic Value of the Hedgehog Signaling Pathway in Ovarian Cancer"

_ijms, 2025, doi:10.3390/ijms26125888_

Round 1

Reviewer 1 Report

Comments and Suggestions for Authors

Salman et al. aimed to investigate the prognostic value of the hedgehog signaling pathway in ovarian cancer, which is highly relevant and timely topic in cancer biology. The study investigates the association between expression levels of hedgehog pathway related genes and clinical outcomes in ovarian cancer (OC). This research contributes valuable insights, particularly in identifying potential gene markers with prognostic and therapeutic implications. However, several suggestions are necessary before possible publication in the journal.

  1. Abstract should be of maximum 200 words, while it exceeds the word limit, please make it more concise.
  2. All paragraphs in introductions should be of equal length to maintain the representation of the manuscript.
  3. Please add citation in the 1st paragraph of introduction
  4. The goals described at the end of the introduction should be marked as a separate paragraph for better knowledge delivery.
  5. “*” in table 1 is representing what?
  6. In section 2.2 (L-16) the “M” after Ct should be in small letter
  7. “Means ± standard errors” written in third last line of section 2.2 is incorrect, please replace it with “Mean ± standard error”
  8. Improve English of first two lines from section 2.3 and in the whole manuscript moderate English improvements are required.
  9. “p” in last line of section 2.3 should be italic.
  10. Please fix the spacing error before section 3.4.
  11. The word “Table 8” should be bold, and this tables should be in correct formatting. And first row of table must be bold and “p” in 2nd column should also be in italic.
  12. The word “Figure 7” should also be bold.
  13. Before discussion, please remove double space.
  14. Please consider suggesting future research directions.
  15. Only 3 recent references (2 from 2022 and 1 from 2023) have been found in the manuscript, please add a few more recent references.
  16. All references should be made following the journal’s reference style.

17. For better scientific communication, the manuscript should be written in a more impersonal and objective tone. 

Comments on the Quality of English Language

The English could be improved to more clearly express the research.

Author Response

First, we would like to thank you for underlining valuable points and suggestions. We appreciate the supportive comment regarding the manuscript, and agree with every single point of the critical points. We accordingly adjusted and revised the manuscript.

Best Regards.

Salman, on behalf of all other authors.

  1. Abstract should be of maximum 200 words, while it exceeds the word limit, please make it more concise. Answer: The Abstract is reduced and made as maximally concise as possible.
  2. All paragraphs in introductions should be of equal length to maintain the representation of the manuscript. Answer: The introduction paragraphs are maintained at approximately equal length.
  3. Please add citation in the 1st paragraph of the introduction. Answer: Citations are now added.
  4. The goals described at the end of the introduction should be marked as a separate paragraph for better knowledge delivery. Answer: The goals of the study are contained as the last paragraph in the introduction.
  5. “*” in table 1 is representing what? Answer: The asterisk is explained in the footnote of the figure; The asterisk indicates the actual number of patients at the public platform Kaplan-Meier Plotter who have OS data and only arrays that passed the quality control. 
  6. In section 2.2 (L-16) the “M” after Ct should be in small letter. Answer: M changed to small letter.
  7. “Means ± standard errors” written in third last line of section 2.2 is incorrect, please replace it with “Mean ± standard error”. Answer: corrected.
  8. Improve English of first two lines from section 2.3 and in the whole manuscript moderate English improvements are required. Answer: The first 2 lines are rephrased. Academic English editing was provided by MDPI Author Services.
  9. “p” in last line of section 2.3 should be italic. Answer: "p" has been changed to italic.
  10. Please fix the spacing error before section 3.4. Answer: Spacing error fixed. To mention, paragraph order has been rearranged for better following of the sections.
  11. The word “Table 8” should be bold, and this tables should be in correct formatting. And first row of table must be bold and “p” in 2nd column should also be in italic. Answer: Table 8, which ist now Table 4, is written in bold, correctly formatted, first row in bold, p in italic.
  12. The word “Figure 7” should also be bold. Figure 7, which is now figure 6, is written in bold.
  13. Before discussion, please remove double space. Answer: Space error corrected.
  14. Please consider suggesting future research directions. Answer: Future direction suggested "As a future direction, a thorough evaluation of SUFU and GLI1 as prognostic biomarkers of high sensitivity and specificity in independent discovery and validation cohorts appears worthwhile. Moreover, our data mark SUFU as a potential therapeutic target that could be evaluated in future preclinical studies. "
  15. Only 3 recent references (2 from 2022 and 1 from 2023) have been found in the manuscript, please add a few more recent references. Answer: We were able to add more recent references (2021-2024) in the introduction and 1 more in the discussion section.
  16. All references should be made following the journal’s reference style. I have issues getting the exact same style order and font with square brackets in the text, I contacted the journal editor for more assistance.

17. For better scientific communication, the manuscript should be written in a more impersonal and objective tone. Answer: This has been widely rephrased in the manuscript to avoid personal language, as possible.

Reviewer 2 Report

Comments and Suggestions for Authors

Major Comments:

  1. Rationale for Focusing on the Hedgehog Pathway - The manuscript lacks a clear rationale for selecting the Hedgehog signaling pathway for prognostic investigation. Why was this particular pathway prioritized over others? This should be explicitly stated in the introduction.
  2. Clarity and Coherence of Writing - The manuscript is poorly structured and difficult to follow. The overall goals of the study and the logical flow between different analyses are often unclear. Additionally, several sections lack proper in-text citation of relevant tables and figures, which makes it difficult in understanding and interpretation for the reader.
    For example, in section 3.1, numerous survival associations are described without clearly referencing the corresponding figure/table (“SUFU, again, was the only member of the HH signaling pathway whose expression demonstrated an adverse prognostic effect on grade 3 OC with HRs of 1.38 and 1.4 and pvalues of 6.3 × 10−5 and 0.007 for PFS and OS, respectively. Moreover, we noticed that the higher expression levels of HHAT, PTCH1, and SUFU were associated with unfavorable PFS, OS, and PFS, respectively, in TP53-mutated OC. SHH, HHAT, PTCH1, and PTCH2 overexpression further improved the survival rates of ovarian cancer patients, particularly when optimal debulking was possible. Interestingly, even in cases where optimal debulking seemed impossible, PTCH2 overexpression was associated with better OS. Once again, SUFU piqued our interest by displaying an adverse correlation with PFS, even in cases with optimal debulking.”).
  3. Gene Selection unclear -   The process used to select Hedgehog pathway genes is not described. Authors should clearly state the database (e.g., KEGG, Reactome, MSigDB) or criteria that was used to extract the Hedgehog pathway-associated gene list.
  4. Data Preprocessing and Normalization -  In the methods section, there is insufficient detail on data preprocessing. The authors need to describe how raw data were normalized and whether batch effect correction was applied—especially critical when merging datasets from different cohorts (as in Table 1). Without such information, the integrity and reproducibility of the analysis are questionable.
  5. Microarray Probe Selection criteria for genes -  The authors mention specific Affymetrix IDs for each gene,  “The Affymetrix IDs that represent our genes are as follows: SHH: 207586_at, HHAT: 219687_at, GLI1: 206646_at, GLI2: 228537_at, GLI3: 227376_at, PTCH1: 209816_at, PTCH2: 221292_at, and SUFU:222749_at.”. But, microarray platforms often include multiple probes per gene. How was the final probe chosen? Was it based on expression average, annotation, or highest signal intensity? This choice should be justified.

Specific Comments on Results Section:

  1. Table 5: The caption is vague and needs clarification. It should concisely describe the content and context of the table. Consider moving detailed notes to footnotes for better readability. The manuscript mentions correlations between gene expression and tumor stage, but the correlation coefficients or statistical values (e.g., p-values or r-values) are not present in the table. These should be clearly included.

  2. Figure 3: The authors should indicate that higher expression of BCNS genes is also associated with better prognosis, as this is a relevant finding.

  3. Section 3.3 – Gene Expression in Cancer Cell Lines: PCR results should be visualized as boxplots with statistical annotations (e.g., p-values). This would more effectively convey differences in gene expression between primary and metastatic lines than the current presentation. Consequently, Table 7 becomes redundant and could be removed if expression patterns are clearly shown graphically.

Minor Comment - Authors should properly cite sentences in the introduction and Discussion sections by providing proper references.

For Example - In the first statement, the authors mention that the overall survival rate is "low" without providing specific statistics or a reference. It would strengthen the introduction if they clearly stated the approximate survival rate (e.g., 5-year overall survival rate) for ovarian cancer and supported it with a credible reference. This would provide context for the clinical relevance of their study.

Comments on the Quality of English Language

The manuscript is poorly structured and difficult to follow. The overall goals of the study and the logical flow between different analyses are often unclear. It difficult in understanding and interpretation for the reader.

Author Response

Thank you very much for your thorough and valuable comments. We agree with the aspects pointed out and have tried to revise them accordingly.  

Major Comments:

  1. Rationale for Focusing on the Hedgehog Pathway - The manuscript lacks a clear rationale for selecting the Hedgehog signaling pathway for prognostic investigation. Why was this particular pathway prioritized over others? This should be explicitly stated in the introduction. Answer: The rationale for selecting the Hedgehog signaling pathway for prognostic investigation is now stated in the last paragraph of the introduction "Building on prior research evaluating the HH pathway in other cancers, we aimed to similarly assess the relevance of the gene expression levels of key members and targets of the HH pathway to patient outcomes by assessing the progression-free and overall survival of OC patients. HH genes (SHH, PTCH1, PTCH2, GLI1, GLI2, GLI3, HHAT, and SUFU) were selected from the KEGG Database, with currently available oncological literature, as some of them have been the subject of previous research, especially the GLI family of transcriptional factors."
  2. Clarity and Coherence of Writing - The manuscript is poorly structured and difficult to follow. The overall goals of the study and the logical flow between different analyses are often unclear. Additionally, several sections lack proper in-text citation of relevant tables and figures, which makes it difficult in understanding and interpretation for the reader.
    For example, in section 3.1, numerous survival associations are described without clearly referencing the corresponding figure/table (“SUFU, again, was the only member of the HH signaling pathway whose expression demonstrated an adverse prognostic effect on grade 3 OC with HRs of 1.38 and 1.4 and pvalues of 6.3 × 10−5 and 0.007 for PFS and OS, respectively. Moreover, we noticed that the higher expression levels of HHAT, PTCH1, and SUFU were associated with unfavorable PFS, OS, and PFS, respectively, in TP53-mutated OC. SHH, HHAT, PTCH1, and PTCH2 overexpression further improved the survival rates of ovarian cancer patients, particularly when optimal debulking was possible. Interestingly, even in cases where optimal debulking seemed impossible, PTCH2 overexpression was associated with better OS. Once again, SUFU piqued our interest by displaying an adverse correlation with PFS, even in cases with optimal debulking.”). Answer: The manuscript has been re-structured and the positions of the sections and paragraphs have been properly organized for better following and a better logical flow between the study designs was improved by stressing the rationale of each design at the beginning of each section. The Tables and Figures are now properly referenced in each relevant text in the result sections. 
  3. Gene Selection unclear -   The process used to select Hedgehog pathway genes is not described. Authors should clearly state the database (e.g., KEGG, Reactome, MSigDB) or criteria that was used to extract the Hedgehog pathway-associated gene list. Answer: the database and criteria are described: "HH genes (SHH, PTCH1, PTCH2, GLI1, GLI2, GLI3, HHAT, and SUFU) were selected from the KEGG Database, with currently available oncological literature, as some of them have been the subject of previous research, especially the GLI family of transcriptional factors."
  4. Data Preprocessing and Normalization -  In the methods section, there is insufficient detail on data preprocessing. The authors need to describe how raw data were normalized and whether batch effect correction was applied—especially critical when merging datasets from different cohorts (as in Table 1). Without such information, the integrity and reproducibility of the analysis are questionable. Answer: stated as "To avoid differences due to different sensitivity, specificity, and dynamic range in detecting gene expression levels for specific genes by different technologies, the search has been narrowed to only tumor samples examined using the in situ oligonucleotide array platforms GPL96 (Affymetrix Human Genome U133A Array), GPL571 (Affymetrix Human Genome U133A 2.0 Array), and GPL570 (Affymetrix Human Genome U133 Plus 2.0 Array). The oligonucleotide gene array files were MAS5 normalized. Then, a second scaling normalization was executed to set the mean expression in each array to 1000. Only the probes present in the GPL96 platform were used in the scaling normalization to prevent platform-specific differences due to the higher probe number in the GPL570 arrays ".
  5. Microarray Probe Selection criteria for genes -  The authors mention specific Affymetrix IDs for each gene,  “The Affymetrix IDs that represent our genes are as follows: SHH: 207586_at, HHAT: 219687_at, GLI1: 206646_at, GLI2: 228537_at, GLI3: 227376_at, PTCH1: 209816_at, PTCH2: 221292_at, and SUFU:222749_at.”. But, microarray platforms often include multiple probes per gene. How was the final probe chosen? Was it based on expression average, annotation, or highest signal intensity? This choice should be justified. Answer: stated as "Only the JetSet best probe set was chosen; this set selects the optimal probe set for each gene using a scoring method established to assess each probe set for specificity, coverage, and degradation".

Specific Comments on Results Section:

  1. Table 5: The caption is vague and needs clarification. It should concisely describe the content and context of the table. Consider moving detailed notes to footnotes for better readability. The manuscript mentions correlations between gene expression and tumor stage, but the correlation coefficients or statistical values (e.g., p-values or r-values) are not present in the table. These should be clearly included. Answer: This table ist now Table 2 and has been extended to include the correlation of expression level of genes in different clinical aspects. Only significant results were indicated in detail. p-value and HR to the stage is listed.

  2. Figure 3: The authors should indicate that higher expression of BCNS genes is also associated with better prognosis, as this is a relevant finding. Answer: BCNS is ID-Affymetrix of PTCH1, and is referred to in the text.

  3. Section 3.3 – Gene Expression in Cancer Cell Lines: PCR results should be visualized as boxplots with statistical annotations (e.g., p-values). This would more effectively convey differences in gene expression between primary and metastatic lines than the current presentation. Consequently, Table 7 becomes redundant and could be removed if expression patterns are clearly shown graphically. Answer: Suggestion is applied and PCR results are demonstrated as suggested.
  4. Minor Comment - Authors should properly cite sentences in the introduction and Discussion sections by providing proper references. Answer: We tried to reference all statements in the introduction, discussion and methodology.

For Example - In the first statement, the authors mention that the overall survival rate is "low" without providing specific statistics or a reference. It would strengthen the introduction if they clearly stated the approximate survival rate (e.g., 5-year overall survival rate) for ovarian cancer and supported it with a credible reference. This would provide context for the clinical relevance of their study.

Comments on the Quality of English Language

The manuscript is poorly structured and difficult to follow. The overall goals of the study and the logical flow between different analyses are often unclear. It is difficult in understanding and interpretation for the reader.

Answer: Academic english editing from MDPI author service was applied. 

The manuscript has been extensively rearranged for better logical and structural following.